# Pharmacodynamics of Ceftriaxone, Ertapenem, Fosfomycin and Gentamicin in *Neisseria gonorrhoeae*

**DOI:** 10.3390/antibiotics11030299

**Published:** 2022-02-23

**Authors:** Urša Gubenšek, Myrthe de Laat, Sunniva Foerster, Anders Boyd, Alje Pieter van Dam

**Affiliations:** 1Department of Infectious Diseases, Public Health Service Amsterdam, Nieuwe Achtergracht 100, 1018 WT Amsterdam, The Netherlands; ugubensek@ggd.amsterdam.nl (U.G.); m.m.delaat@amsterdamumc.nl (M.d.L.); sunniva.foerster@gmail.com (S.F.); aboyd@ggd.amsterdam.nl (A.B.); 2Stichting HIV Monitoring—The Dutch HIV Monitoring Foundation, Nicolaes Tulphuis, Tafelbergweg 51, 1105 BD Amsterdam, The Netherlands; 3Department of Medical Microbiology, Amsterdam University Medical Center, Meibergdreef 9, 1105 AZ Amsterdam, The Netherlands

**Keywords:** *Neisseria gonorrhoeae*, antimicrobial resistance, time–kill curves, pharmacodynamics, ceftriaxone, ertapenem, fosfomycin, gentamicin

## Abstract

Objectives: To assess the in vitro effect of select antimicrobials on the growth of *N. gonorrhoeae* and its pharmacodynamic parameters. Methods: Time–kill assays were performed on two reference *N. gonorrhoeae* strains (ceftriaxone-resistant WHO X and ceftriaxone-susceptible WHO F) and one clinical *N. gonorrhoeae* strain (ceftriaxone-susceptible CS03307). Time–kill curves were constructed for each strain by measuring bacterial growth rates at doubling antimicrobial concentrations of ceftriaxone, ertapenem, fosfomycin and gentamicin. Inputs from these curves were used to estimate minimal bacterial growth rates at high antimicrobial concentrations (*ψ*_min_), maximum bacterial growth rates in the absence of antimicrobials (*ψ*_max_), pharmacodynamic minimum inhibitory concentrations (zMIC), and Hill’s coefficients (κ). Results: Ceftriaxone, ertapenem and fosfomycin showed gradual death overtime at higher antimicrobial concentrations with a relatively high *ψ*_min_, demonstrating time-dependent activity. Compared to WHO F, the *ψ*_min_ for WHO X was significantly increased, reflecting decreased killing activity for ceftriaxone, ertapenem and fosfomycin. At high ceftriaxone concentrations, WHO X was still efficiently killed. CS03307 also showed a high *ψ*_min_ for ceftriaxone in spite of a low MIC and no difference in *ψ*_min_ for fosfomycin in spite of significant MIC and zMIC differences. Gentamicin showed rapid killing for all three strains at high concentrations, demonstrating concentration-dependent activity. Conclusions: Based on time–kill assays, high-dosage ceftriaxone could be used to treat *N. gonorrhoeae* strains with MIC above breakpoint, with gentamicin as a potential alternative. Whether ertapenem or fosfomycin would be effective to treat strains with a high MIC to ceftriaxone is questionable.

## 1. Introduction

Ceftriaxone is recommended for routine treatment of gonorrhea based on guidelines from the United States, United Kingdom and Europe [1,2,3]. Meanwhile, there have been multiple reports of *N. gonorrhoeae* (*Ng*) strains exhibiting resistance to ceftriaxone, which has resulted in treatment failure [4,5,6]. These reports are concerning, as outbreaks of ceftriaxone-resistant *Ng* could make antimicrobial-resistant *Ng* a serious public health threat. At present, ceftriaxone is the only option to treat gonorrhea with a single antibiotic dose. Moreover, gonorrhea is usually diagnosed with a nucleic acid amplification test which does not provide results regarding resistance. Therefore, treatment of gonorrhea is now becoming problematic since ceftriaxone-resistant strains have emerged. This has become more relevant since guidelines, such as those mentioned above, have omitted to recommend combination therapy of ceftriaxone and azithromycin, and now ceftriaxone monotherapy is again recommended in guidelines [1,2]. There is, then, an urgent need to evaluate other antimicrobials that could be used as first-line treatment. Ertapenem, gentamicin and fosfomycin have been identified as potential candidates to treat antimicrobial-resistant *Ng* [7].

Establishing clinical use for potential antimicrobials requires information regarding their pharmacokinetic (PK) and pharmacodynamic (PD) properties. PK/PD indices, such as the maximal time above the minimum inhibitory concentration (i.e., ƒT > MIC), integrated minimum inhibitory concentration (i.e., ƒAUC/MIC) or maximum concentration of drug in serum-to-minimum inhibitory concentration (i.e., ƒC_max_/MIC) are useful in approximating the appropriate doses of antimicrobials and can be used to do so far more accurately than the minimum inhibitory concentration (MIC) alone [8,9,10]. In vitro assays, such as time–kill assays [TKA), can also be used to estimate other parameters from PK/PD models [11,12], such as the minimal bacterial growth rate at high antimicrobial concentrations (*ψ*_min_), maximum bacterial growth rate in the absence of antimicrobials (*ψ*_max_), pharmacodynamic MICs (zMIC) and Hill coefficient (κ) [13], which provide further insight into the relationship between bacterial growth (which can be negative in the presence of antibiotics), antimicrobial concentrations and the rapidity of bacterial killing in the presence of sufficient concentrations of antibiotics [10].

Multiple in vitro studies have already examined the PD parameters of ceftriaxone and gentamicin on ceftriaxone-susceptible strains of *Ng* [9,14,15,16,17,18]. However, PD parameters for ertapenem and fosfomycin are still lacking for *Ng*, with fosfomycin known to exhibit wide variation in PD parameters across organisms [19]. PD parameters are also largely unknown for ceftriaxone-resistant strains of *Ng*. No MIC breakpoints for *Ng* have been determined for ertapenem, gentamicin and fosfomycin, making it difficult to define resistance.

The purpose of the present research was to study the in vitro effect of four antimicrobials, recently evaluated in a randomized clinical trial (i.e., ceftriaxone, ertapenem, fosfomycin and gentamicin) [7], on the rate of killing of *Ng*. We additionally established PD parameters for strains differing in susceptibility to ceftriaxone and fosfomycin. The purpose of the research was also to assess the effectivity of treatment of infections with a ceftriaxone-resistant *Ng* strain with the antibiotics under study.

## 2. Results

### 2.1. Growth of N. gonorrhoeae

Growth of all three strains (WHO F, WHO X and CS03307) was well supported in modified GW medium. For all strains, log growth phase started to occur after 4 h of incubation (*t* = 0). Of note, the starting inoculum for the WHO X strain had to be increased by 10 times those of the other strains in order to reach the log phase after 4 h of incubation.

### 2.2. Time–Kill Curves

Selected time–kill curves (TKC) for the WHO F, WHO X and CS03307 strains using ceftriaxone, ertapenem, fosfomycin and gentamicin are shown in Figure 1, Figure 2, Figure 3 and Figure 4. The other two experiments used for the calculations are shown in the Appendix A (Appendix A). Differences between experimental replicates were taken into account when calculating farmacodynamic parameters, as described in the next section.

Ceftriaxone induced gradual death over time for all three strains at concentrations of 0.5× MIC and higher. The WHO F strain showed the most dramatic decrease in growth (Figure 1A). The ceftriaxone-resistant strain WHO X was also steadily killed in the presence of 0.75 mg/l ceftriaxone (Figure 1B), although the decrease over time was much less pronounced compared to the WHO F or CS03307 strains (Figure 1B,C, respectively).

At the MIC of ertapenem, the number of CFU/ml over time remained constant for the WHO F (Figure 2A), WHO X (Figure 2B) and CS03307 (Figure 2C) strains, while ertapenem concentrations above the MIC were necessary to induce killing for all three strains. Although the WHO X strain had a rather low MIC of 0.032 mg/l, it was not efficiently killed by ertapenem, even at higher concentrations (Figure 2B).

Concentrations above the MIC for fosfomycin were necessary to induce killing of the WHO F (Figure 3A), WHO X (Figure 3B) and CS03307 (Figure 3C) strains. Similar to ertapenem, fosfomycin, even at high concentrations, was not efficient in killing the WHO X strain (Figure 3B). The strain CS03307, selected for its low MIC to fosfomycin, was indeed killed at lower fosfomycin concentrations (Figure 3C) compared to WHO F (Figure 3A), although the kinetics of the killing rates appeared otherwise similar.

For gentamicin, concentrations above the MIC were also required to induce killing of the WHO F (Figure 4A), WHO X (Figure 4B) and CS03307 (Figure 4C) strains. Rapid killing to below the limit of detection (200 CFU/mL) was achieved in less than three hours at the highest two to three concentrations of gentamicin for all three strains (WHO F, Figure 4A; WHO X, Figure 4B; CS03307, Figure 4C).

### 2.3. Pharmacodynamic Parameters

The average maximal growth rate (*ψ*_max_) across all strains was 0.62 h^−1^ (95% confidence interval (CI): 0.55–0.70 h^−1^), which corresponds to a bacterial doubling time of (*t*_1/2_) 66 min (95%CI: 59–75 min).

The estimated zMIC values generally corresponded well with MIC, as measured by the Etest (within one to two doubling dilutions) (Table 1). The only exception was ceftriaxone, which had a markedly lower zMIC than MIC.

The minimal bacterial growth rate at high antimicrobial concentrations (*ψ*_min_) is shown as the lower, horizontal asymptote in Figure 5A (ceftriaxone), Figure 5B (ertapenem), Figure 5C (fosfomycin) and Figure 5D (gentamicin). Compared to the WHO X and CS03307 strains, the *ψ*_min_ values for WHO F were significantly lower for ceftriaxone (*p* < 0.0001 and *p* < 0.0001, respectively) and ertapenem (*p* < 0.0001 and *p* < 0.0001, respectively). In addition, *ψ*_min_ values for the ceftriaxone-resistant strain WHO X were significantly higher than the clinical strain CS03307 for ertapenem (*p* = 0.0049), reflecting the slower killing of WHO X in vitro at the highest concentrations of this antibiotic. *ψ*_min_ values were also significantly lower for WHO F than WHO X regarding fosfomycin (*p* < 0.0001). No significant differences in *ψ*_min_ were observed between strains for gentamicin; however, estimation of *ψ*_min_ might not have been robust given that the lower asymptote at higher antibiotic concentrations could not be clearly estimated for some experiments.

The Hill coefficient, *κ*, for ceftriaxone was significantly lower for the WHO F strain in comparison to the CS03307 strain (*p* < 0.0001), reflecting a steeper increase of killing activity around zMIC levels for this strain (Figure 5A). Differences in *κ* were also observed for ertapenem between the WHO F and CS03307 versus WHO X strains (*p* = 0.011 and *p* = 0.0013, respectively). For all other antibiotics, no significant differences in *κ* were observed between strains.

## 3. Discussion

With the emergence of antimicrobial-resistant strains of *Ng,* especially those resistant to ceftriaxone, there is an urgent need to evaluate new treatment options for *Ng* infections. A recent clinical trial comparing the efficacy of gentamicin, ertapenem and fosfomycin versus ceftriaxone [7] demonstrated non-inferiority for intramuscularly administered ertapenem 1 g to intramuscularly administered ceftriaxone 500 mg. Non-inferiority could not, however, be shown for gentamicin 5 mg/kg (intramuscularly administered) or fosfomycin 6 g (orally administered). In the present study, the pharmacodynamic characteristics of these antimicrobials have been compared, specifically using a well-described time–kill assay [9] that evaluates the relationship between antimicrobial concentrations and bacterial killing rates [20].

In our time–kill assays, we included two reference strains, one ceftriaxone-susceptible (WHO F) and the other ceftriaxone-resistant (WHO X), as well as a clinical strain that was susceptible to ceftriaxone but with a slightly increased MIC for ceftriaxone and a lower MIC for fosfomycin (CS03307). Of note, the WHO X strain also had a somewhat higher MIC against ertapenem. The growth of all strains was well supported in growth medium [9], allowing for a 2-log increase in CFU/ml within 10 h. To reach the same growth rate of the WHO F and CS03307 strains, we had to use a 10-times higher quantity of inoculum of the WHO X compared to the other two strains. WHO X colonies were also visibly smaller, which might be due to the lowered fitness of strains with *penA* mutations [21,22] in vitro. This finding could also explain why resistant strains are transmitted less frequently from infected sex partners [23].

In the absence of antimicrobials, all strains showed consistent growth characteristics, as demonstrated by their comparable *ψ*_max_ values. The WHO F strain, which was highly susceptible to ceftriaxone, was much more quickly killed than the other two strains in the presence of ceftriaxone, while the killing rates obtained in our study were faster compared to those reported by others [9]. Interestingly, the *ψ*_min_ did not differ between the CS03307 (ceftriaxone-susceptible) and WHO X (ceftriaxone-resistant) strains, indicating that a higher dose of ceftriaxone, if clinically feasible, could be sufficient to eliminate strains with an MIC immediately over the clinical breakpoint.

The pharmacodynamic parameter zMIC reflects the antimicrobial concentration that is required to prohibit growth, as estimated from this in vitro assay, and was for ceftriaxone, in our case, roughly three-fold lower than the empirical MIC. Our data suggest that for efficient killing of a ceftriaxone-resistant strain like WHO X a desirable, unbound plasma ceftriaxone concentration would be 2–3 mg/L after 24 h which would require twice-daily administration of 2 g [24]. Currently, the highest doses of ceftriaxone recommended during first-line treatment do not exceed 1 g as a single dose [25], which results in an unbound plasma concentration of 0.7 mg/l after 24 h [24]. This concentration would be insufficient to treat ceftriaxone-resistant *Ng* infections. A recent study also demonstrated 100% clearance rates of *Ng*, including ceftriaxone-resistant *Ng*, when using higher doses of ceftriaxone (i.e., greater than 1 g in a single dose) [26]. Taken together, an additional clinical breakpoint for high- and multidose treatment of *Ng* with ceftriaxone could be established.

The PK/PD characteristics of ertapenem have not been previously evaluated with this assay. The *ψ*_min_ values were lower for this antibiotic compared to ceftriaxone, at least for both WHO strains, thus suggesting less efficient killing. The WHO X strain also demonstrated significantly higher *ψ*_min_ for ertapenem than the WHO F and CS03307 strains. Given that the WHO X strain also had much a higher zMIC and MIC, the increased MIC for ertapenem also likely reflects diminished killing rates. In contrast to ceftriaxone, no noticeable departures between MIC and zMIC were found for all strains. These findings make it questionable whether ertapenem would be a good alternative to treat ceftriaxone-resistant *Ng* strains. Until now, MIC values for ertapenem have varied between 0.008 and 0.5 mg/L [16,27]. No strains of *Ng* above the EUCAST non-species related PK/PD breakpoint (i.e., 0.5 mg/L) have been observed. However, this breakpoint is based on standard dosing and not single-dose treatment, the latter of which would be intended for *Ng* infections. It would be valuable to perform TKAs in strains of *Ng* with a higher MIC of ertapenem.

Fosfomycin, whose use has been limited in clinical trials, is effective against *Ng*, but not as a single high-dose treatment [7]. The two WHO reference strains had similar MIC and zMIC for fosfomycin, whereas the clinical strain CS03307, purposely by our a priori selection, had a lower MIC and zMIC., A single, orally administered 3 g dose of fosfomycin, which reflects a maximum serum concentration (C_max_) of 27 ± 6 mg/L [28], may be sufficient to eliminate this strain, but is unlikely to clear WHO F and WHO X strains, considering their high MIC and zMIC.

We observed similar pharmacodynamics for gentamicin, as described previously [9]. This antibiotic exhibited a quick and strong bactericidal effect on all three strains, with *ψ*_min_ values much lower compared to the other antimicrobials tested. The ceftriaxone-resistant strain WHO X demonstrated similar killing rates as compared to the ceftriaxone-susceptible WHO F and CS03307. Gentamicin could then be considered as a salvage treatment for ceftriaxone-resistant strains. However, clinical trials evaluating the use of this antibiotic on ceftriaxone-susceptible strains have failed to demonstrate non-inferior efficacy to ceftriaxone [7,18]. Gentamicin is unlikely to be an optimal treatment for gonorrhea. Based on our time–kill curves, gentamicin should be expected to perform better as a treatment option than the results found in the clinical trials cited above. One major reason to explain treatment failures of gentamicin could be that gentamicin does not reach appropriate intracellular concentrations, and that intracellular survival of gonococci might occur [29], especially after single-dose administration.

MIC are easy to obtain and are, therefore, often regarded as a good approximation of the efficacy of an antibiotic therapy. Studies such as ours show that other parameters could also be measured and might be more representative of treatment efficacy. Pharmacodynamic parameters could also be used as predictors of PK/PD indices. For example, antimicrobials whose killing action is time-dependent are associated with higher *ψ*_min_ values, whereas those whose killing action is concentration-dependent have lower *ψ*_min_ values [9,30]. The higher *ψ*_min_ values of ceftriaxone, ertapenem and fosfomycin would suggest that killing against *Ng* is time-dependent and the lower *ψ*_min_ values of gentamicin that it is concentration-dependent. In previous studies, κ, which describes the steepness of the pharmacodynamic curve around the zMIC, has been commonly used to assess the working mechanisms of antimicrobials. Regoes [20] hypothesized that high and low values of κ are associated with concentration- and time-dependent antimicrobials, respectively. Since the κ values were not much different between time- and concentration-dependent antimicrobials, as defined by the *ψ*_min_ values, this hypothesis does not seem to hold, as was observed by others [9]. Using recent improvements of this type of assay [31], it could well be that these parameters might be more easily measured in the near future.

A major limitation of the TKA is the lengthy processing time needed to complete each experiment. Due to these constraints, we chose a 6 h timeframe. Consequently, it was entirely possible that some antimicrobial effects were unable to be assessed (e.g., other *ψ*_min_ values could have been obtained for gentamicin after longer incubation periods). Further studies would be needed to clarify the kinetics of possible regrowth and/or patterns of post-antibiotic effects.

One limitation of the method could be that we used 96-well plates for the incubations in TKA and took samples over time from different wells. In replicate wells, growth of gonococci might differ. We did this in order to take samples from cultures with different antibiotics and different concentrations of these antibiotics rapidly and exactly at the same moment. Using multiwell plates or tubes to take samples would have taken substantially more time but the advantage could have been that exactly the same cultures would have been studied. We could study many replicates at *t* = 0 and usually a twofold difference in cfu/ml between wells was maximally found. These differences are much lower than the 10–10,000-fold differences in dilutions after adding antibiotics.

Another limitation of the study is the fact that antimicrobial efficacy is highly dependent on the media conditions selected for testing. This is not always directly translatable to in vivo conditions. As stated above, in vivo pathways for gonococcal survival in the presence of antibiotics, like persistence in intracellular sites, cannot be studied by time–kill assays.

## 4. Conclusions

Based on TKA, we demonstrate that all four antimicrobials (i.e., ceftriaxone, ertapenem, fosfomycin and gentamicin) are effective against ceftriaxone-susceptible *Ng* strains. Gentamicin and ceftriaxone, at higher doses, are further effective against a ceftriaxone-resistant strain of *Ng*. Bacterial killing of this strain is less effective when using ertapenem and fosfomycin at concentrations above the MIC. Obtaining clinical pharmacokinetic data is encouraged to support the pharmacodynamic parameters obtained in this study, as this information could help clarify the role of these antibiotics in a clinical setting.

## 5. Methods

### 5.1. Neisseria Gonorrhoeae Strains and Media

To ensure a wide variation in MIC, we chose three *Ng* strains based on their MIC to ceftriaxone. Two international *Ng* reference strains, one a ceftriaxone-susceptible strain, WHO F (origin: Canada, 1991), with a wild type *penA* allele [32], and the other a ceftriaxone-resistant strain, WHO X, with a *penA* mosaic allele (H041; Japan, 2009) [5], and a clinical strain CS03307 with a low MIC for fosfomycin were included in this study. For the clinical strain, gonococcal species verification was performed, including growth on selective BD^TM^ GC-Lect^TM^ Agar (BD, Drachten, The Netherlands), oxidase (BD, The Netherlands) and catalase (Merck, Amsterdam, The Netherlands) tests and species identification with MALDI-TOF (Bruker, Leiderdorp, The Netherlands). After identification, a frozen stock (Pro-lab diagnostics, Round Rock, TX, USA) was prepared and stored at –80 °C. All strains were cultured from frozen stocks (–80 °C), on CHOC-GC agar (bioTRADING, Mijdrecht, The Netherlands) for 20–24 h at 37 °C in a 5 % CO_2_-enriched atmosphere. Colonies were subcultured once more on CHOC-GC agar for 24 h under the same conditions for all experiments. For the growth curve and time–kill assays, cultures were transferred to liquid sterile GW medium, which was prepared as described previously [33], with additional 1% ACES (N-(2-Acetamido)-2-aminoethanesulfonic acid) buffer to ensure pH stabilization at a pH of 6.8.

### 5.2. Antimicrobial Susceptibility Testing

MIC was determined for ceftriaxone, ertapenem, fosfomycin and gentamicin for all three strains using the Etest method (AB bioMérieux, Askim, Sweden) according to the manufacturer’s instructions. Empirical MIC values were used to prepare dilutions for the time–kill assays.

### 5.3. Viable Cell Counts

Bacterial viability testing was performed following the method previously described by Foerster et al. [9]. Briefly, bacterial solution was incubated in a 96-well plate. At each time point, 100 µL aliquots, which amounts to the complete volume of the wells to be tested, were transferred to another plate. Subsequently, cultures were serially diluted in seven subsequent 1:10 dilutions (20 µL culture in 180µL sterile phosphate-buffered saline (PBS) (Gibco, Lougborough, UK). Five µL droplets of every dilution were spotted on pre-dried CHOC-GC agar plates and incubated at 37 °C in a 5 % CO_2_-enriched atmosphere. The density of viable bacteria, measured in colony forming units per ml (CFU/mL), was calculated from the first dilution that resulted in a countable range of 3–30 colonies.

### 5.4. Growth Curves

Growth curves were drawn to ensure that *Ng* was at a stable early-to-mid log phase at the time of antimicrobial introduction using a previously described method [9]. However, we shook growing bacteria at 80 rpm at 37 °C in a 5% CO_2_-enriched atmosphere and viable cell counts were made at 0, 4, 5, 6, 7, 8, 9 and 10 h.

### 5.5. Time–Kill Assay

We used the method previously described by Foerster et al. [9], adapted as follows. *N. gonorrhoeae* strains grew in liquid GW medium [33], separately incubated with different antimicrobials (ceftriaxone, ertapenem, fosfomycin and gentamicin). For each antimicrobial, 11 different concentrations were used in a two-fold serial dilution, ranging from 0.016× MIC to 16× MIC and one negative control of PBS. The antimicrobials examined were ceftriaxone (Sigma-Aldrich, Amsterdam, The Netherlands) ertapenem (Sigma-Aldrich, The Netherlands), fosfomycin (Sigma-Aldrich, The Netherlands) and gentamicin (Sigma-Aldrich, The Netherlands). For all assays, CHOC-GC agar plates were used and growing cultures were incubated at 37 °C in a 5% CO_2_-enriched atmosphere. During growth in adapted liquid GW medium, the plates were put on a shaker at 80 rpm under the same conditions. From an overnight culture, a suspension of *Ng* in PBS amounting to a 0.5 McFarland standard was prepared. For the strains WHO F and CS03307, 450 µL of the culture was diluted in 45 mL of pre-warmed (37 °C) GW medium, while for the WHO X strain, 4.5 mL of culture was used in the same volume of GW medium. Diluted bacterial solution was transferred to a reagent reservoir and 90 µL was transferred with an electronic multichannel pipette to each well in round-bottom 96-well Sarstedt microtiter plates. The growth plates were pre-incubated for four hours to ensure stable growth. At time point zero, 10 µL of all 11 antimicrobial concentrations (or PBS) was added to the 90 µL growing cultures, resulting in eight identical rows with bacteria exposed to 11 different antimicrobial concentrations and an untreated control. At chosen time points (−4, 0, 1, 2, 3, 4, 5, 6 h of incubation), 100 µL samples from all wells in one of the eight rows were taken, serially diluted and used for viable cell count determination, as described above. Experiments were repeated at least three times. Experiments in which the growth control (no added antibiotics) showed more than a one log decrease in the number of CFU/ml were regarded as non-valid because of bad growth conditions for the strain. Experiments in which bacterial killing was not evident after 5 h of incubation at the highest antimicrobial concentration were also regarded as non-valid because antibiotics were unlikely to have been administered correctly. The first three successful experiments that met these requirements were considered in analysis.

### 5.6. Estimating Bacterial Growth Rates

Bacterial growth rates (*ψ)* were determined from changes in CFU/ml during the first 6 h of the time–kill assays with a detection limit of 200 CFU/ml. The function of total bacterial density over time, $N\left(t\right)$, allowed for growth or death at a constant rate from baseline levels, $N_{0}$, resulting in an exponential increase or decrease, respectively, in bacterial density:(1)$$N\left(t\right)=N_{0}\cdot e^{\psi t}$$

Growth rates were estimated by regressing the natural logarithm of colony counts on time as a linear function. We used maximum likelihood estimation to obtain parameter estimates from this model, while accounting for bacterial counts below the limit of detection (i.e., censored). For a given antimicrobial, the geometric mean of all measurements at zero hours was used as the first data point. Bacterial doubling time can be calculated from the growth rate as follows:(2)$$T_{\frac{1}{2}}=\frac{\ln\left(2\right)}{\psi }.$$

### 5.7. Pharmacodynamic Model

We used the pharmacodynamic model proposed by Regoes et al. [20], in which the relationship between bacterial growth rates (*ψ)* and the concentration of an antimicrobial (a) is defined as:(3)$$\psi \left(a\right)=\psi_{max}-\frac{\;\left(\psi_{max}-\psi_{min}\right){\left(\frac{a}{zMIC}\right)}^{\kappa }}{{\left(\frac{a}{zMIC}\right)}^{\kappa }-\frac{\psi_{min}}{\psi_{max}}},\;$$
where *ψ*_max_ is the maximal bacterial growth rate in the absence of the antimicrobial and *ψ*_min_ is the minimal bacterial growth rate at high concentrations of the antimicrobial. zMIC is the pharmacodynamic MIC where the bacterial growth rate is zero (i.e., $\psi \left(zMIC\right)=0$). κ denotes the Hill coefficient, which describes the steepness of the reverse sigmoid relationship of $\psi \left(a\right)$.

We estimated the four parameters of the pharmacodynamic model using a system of seemingly unrelated, non-linear regression models. Briefly, we used data from experiments run in triplicate to model parameters according to bacterial strains (WHO F, WHO X and CS03307) for a total of three models. Models were fit with the residual sums of squares methods, while jointly modeling error terms across within-strain models. Parameters were then compared between strains using a Wald χ^2^ test. This procedure was repeated for each antimicrobial (ceftriaxone, ertapenem, fosfomycin and gentamicin).

Statistical significance was defined by a *p*-value < 0.05/3 (i.e., ~0.0167, corrected for multiple comparisons using the Bonferroni method). Statistical analysis was carried out using R (v3.6.3, Vienna, Austria) and STATA (v15.1, College Station, TX, USA).

## Figures and Tables

**Figure 1 antibiotics-11-00299-f001:**
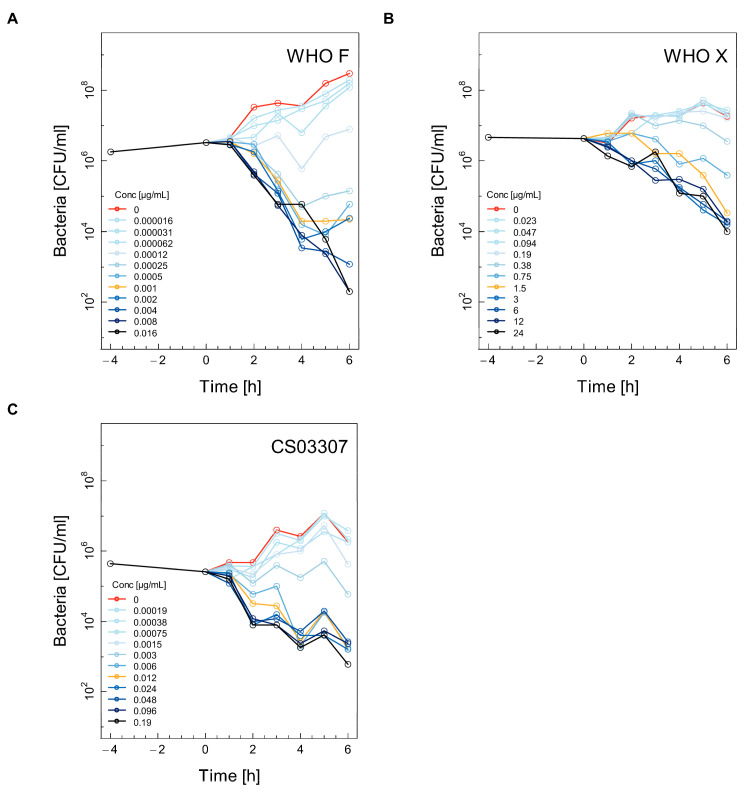
Selected time–kill curves (TKC) for three different strains of *Neisseria gonorrhoeae* using ceftriaxone. For each combination of antibiotic and strain, the TKC from one of the three independent experiments is shown: WHO F (**A**), WHO X (**B**), CS03307 (**C**). For each figure, eleven doubling dilutions are plotted. The black line corresponds to the highest concentration of antibiotics used in the assay (16× the minimum inhibitory concentration (MIC)). The yellow line represents the concentration corresponding to 1× MIC, while the red line represents growth in the absence of antimicrobials. The number of colony-forming units (CFU)/ml was measured from 4 h before until 6 h after the addition of antimicrobials. The limit of detection was 200 CFU/mL.

**Figure 2 antibiotics-11-00299-f002:**
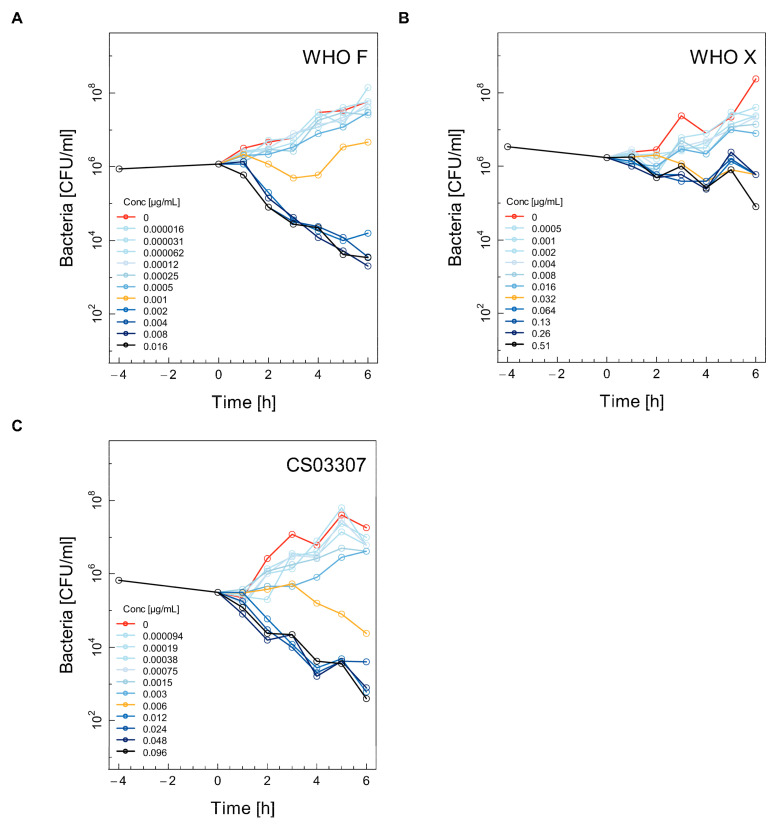
Selected time–kill curves (TKC) for three different strains of *Neisseria gonorrhoeae* using ertapenem. For each combination of antibiotic and strain, the TKC from one of the three independent experiments is shown: WHO F (**A**), WHO X (**B**), CS03307 (**C**). See the legend for Figure 1 for further explanation.

**Figure 3 antibiotics-11-00299-f003:**
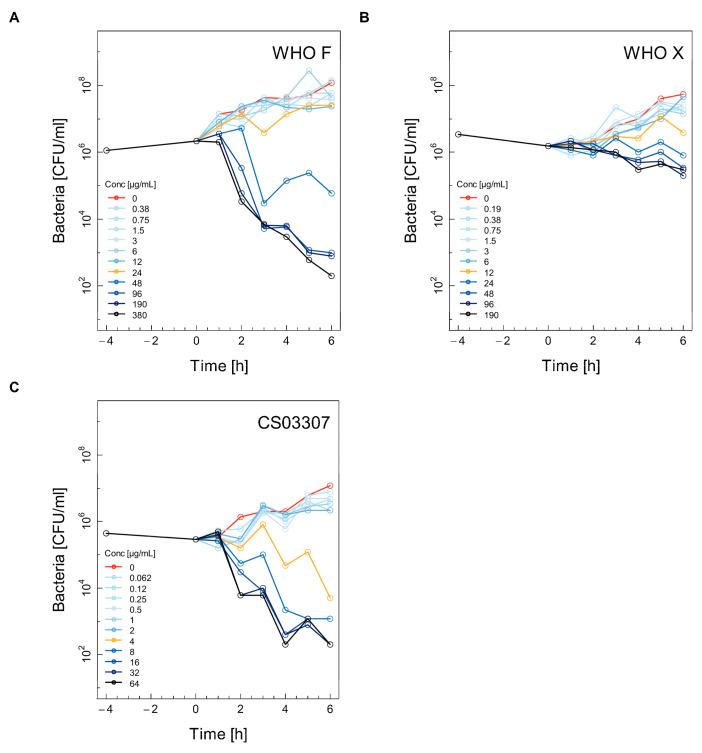
Selected time–kill curves (TKC) for three different strains of *Neisseria gonorrhoeae* using fosfomycin. For each combination of antibiotic and strain, the TKC from one of the three independent experiments is shown: WHO F (**A**), WHO X (**B**), CS03307 (**C**). See the legend for Figure 1 for further explanation.

**Figure 4 antibiotics-11-00299-f004:**
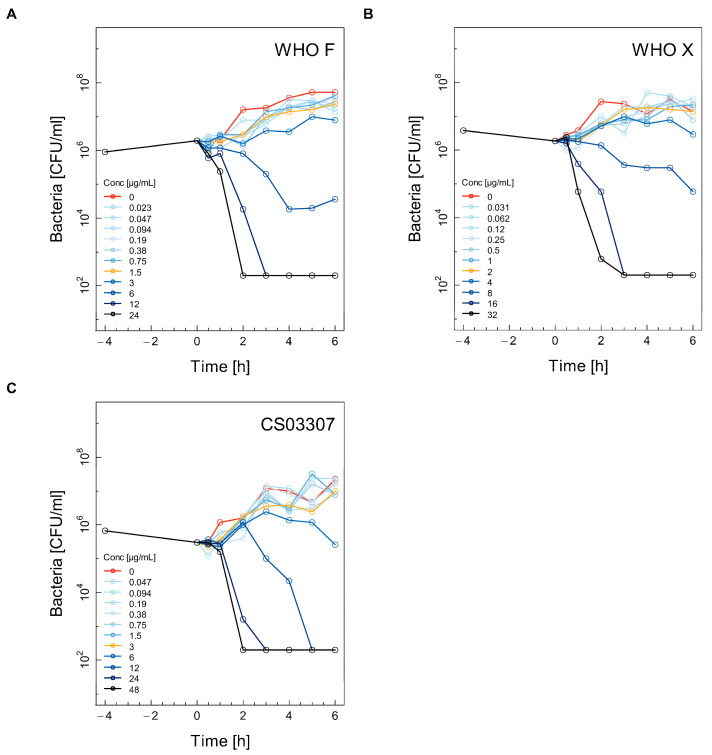
Selected time–kill curves (TKC) for three different strains of *Neisseria gonorrhoeae* using gentamicin. For each combination of antibiotic and strain, the TKC from one of the three independent experiments is shown: WHO F (**A**), WHO X (**B**), CS03307 (**C**). See the legend for Figure 1 for further explanation.

**Figure 5 antibiotics-11-00299-f005:**
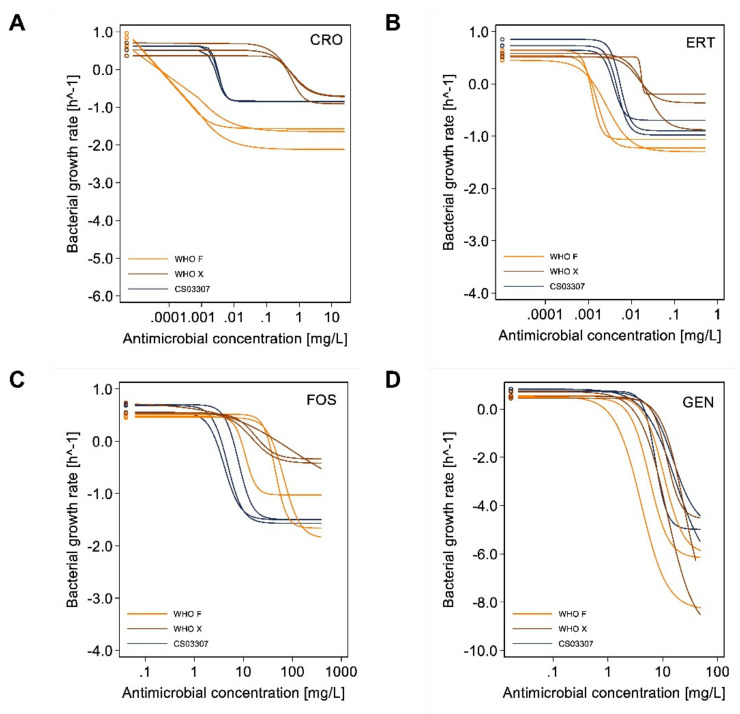
Curves representing the pharmacodynamic functions of specific antimicrobials (i.e., ceftriaxone (**A**), ertapenem (**B**), fosfomycin (**C**) and gentamicin (**D**)) for the three different strains of *Neisseria gonorrhoeae*. Curves for the WHO F strain (ceftriaxone susceptible, high fosfomycin MIC) are presented in yellow lines, WHO X (ceftriaxone resistant, high fosfomycin MIC) in brown lines and CS03307 (ceftriaxone susceptible, low fosfomycin MIC) in navy blue lines. Curves from the three independent experiments with each combination of antibiotic and strain are presented. Abbreviations: CRO, ceftriaxone; ERT, ertapenem; FOS, fosfomycin; GEN, gentamicin.

**Table 1 antibiotics-11-00299-t001:** Pharmacodynamic parameters estimated for each combination of strain and antibiotic.

Antibiotic	Parameter (95%CI)	Strain		
		WHOF	WHOX	CS03307
Ceftriaxone	*ψ* _max_	0.79 (0.40, 1.19)	0.54 (0.44, 0.63)	0.59 (0.54, 0.64)
	*ψ* _min_	−2.00 (−2.44, −1.55)	−0.76 (−0.84, −0.68)	−0.84 (−0.89, −0.79)
	Κ	0.78 (0.41, 1.16)	1.70 (1.13, 2.28)	3.94 (2.60, 5.29)
	zMIC (µg/mL)	0.00012 (0.00007, 0.00017)	0.42 (0.34, 0.49)	0.0029 (0.0027, 0.0031)
	MIC (µg/mL)	<0.002	1.5	0.012
Ertapenem	*ψ* _max_	0.55 (0.49, 0.61)	0.61 (0.48, 0.73)	0.74 (0.67, 0.81)
	*ψ* _min_	−1.13 (−1.23, −1.02)	−0.54 (−0.72, −0.37)	−0.85 (−0.94, −0.76)
	Κ	2.60 (1.85, 3.35)	1.26 (0.56, 1.96)	3.78 (2.47, 5.06)
	zMIC (µg/mL)	0.0012 (0.0010, 0.0013)	0.020 (0.013, 0.026)	0.0045 (0.0040, 0.0051)
	MIC (µg/mL)	<0.002	0.032	0.006
Fosfomycin	*ψ* _max_	0.54 (0.39, 0.72)	0.58 (0.52, 0.64)	0.66 (0.57, 0.75)
	*ψ* _min_	−1.66 (−2.14, −1.18)	−0.37 (−0.48, −0.25)	−1.55 (−1.74, −1.36)
	Κ	1.36 (0.58, 2.15)	1.73 (1.00, 2.47)	2.31 (1.54, 3.08)
	zMIC (µg/mL)	17.4 (10.5, 24.3)	24.5 (18.9, 30.1)	3.79 (3.17, 4.40)
	MIC (µg/mL)	24	12	4
Gentamicin	*ψ* _max_	0.50 (0.09, 0.92)	0.57 (0.28, 0.86)	0.80 (0.59, 1.01)
	*ψ* _min_	−6.86 (−8.71, −5.00)	−7.91 (−11.69, −4.14)	−5.44 (−6.36, −4.51)
	Κ	1.92 (0.76, 3.08)	1.82 (0.81, 2.84)	1.86 (1.24, 2.49)
	zMIC (µg/mL)	1.6 (0.6, 2.6)	3.9 (2.0, 5.9)	4.2 (3.0, 5.4)
	MIC (µg/mL)	1.5	2	3

Parameter estimates and 95% confidence intervals were obtained using a system of seemingly unrelated, non-linear regression models (Methods). *p*-values comparing parameters between strains are given in the Appendix A (Appendix A).

## Data Availability

The data used and/or analysed during the current study are available from the corresponding author on reasonable request.

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
