# Peer review of "Pharmacodynamics of Ceftriaxone, Ertapenem, Fosfomycin and Gentamicin in Neisseria gonorrhoeae"

_antibiotics, 2022, doi:10.3390/antibiotics11030299_

Round 1

Reviewer 1 Report

This paper addresses the important issue of Neisseria resistance to antibiotics. However, the manuscript needs a thorough rewrite, in particular the “Results” section. The results are presented in a very brief form and sloppily written. Figures should also be drawn again, it are unclear and not very informative. In general, the entire manuscript is not written neatly.

Author Response

Comments and Suggestions for Authors

This paper addresses the important issue of Neisseria resistance to antibiotics. However, the manuscript needs a thorough rewrite, in particular the “Results” section. The results are presented in a very brief form and sloppily written. Figures should also be drawn again, it are unclear and not very informative. In general, the entire manuscript is not written neatly.

We have gone once again through the manuscript and have completely rewritten the Introduction, Results and Discussion section and made a number of improvements in the Methods section. The Results section now describes the findings in a more informative way. We have also made a number of alterations to figure 1. We now describe clearly what is shown in a-l (see also revised legend) and emphasized more strongly the black, yellow and red lines in the figure. Otherwise, the figures are similar to those as presented in earlier manuscript by one of our co-authors (Foerster et al., PMC5027106). The referee does not provide specific suggestion in which way the figures should be redrawn. We hope the figures now meet the standard for publication in the Journal.

Reviewer 2 Report

Thank you for giving me the opportunity to review this paper. It could be an interesting addition to the available knowledge but the way paper is written leaves much questions unanswered. Also, the whole paper needs to be edited to fit the journal requirements regarding formatting. English needs to be carefully checked.

Some specific comments:

Introduction lacks more information on why the present study was warranted. The issue of resistance in NG is not sufficiently described, the issues with available treatment, role ceftriaxone plays etc needs to be emphasized.

We established PD parameters for each antibiotic in a ceftriaxone-susceptible strain (WHO F), ceftriaxone-resistant strain (WHO X), and a non-reference clinical strain CS03307, which were used to identify whether antimicrobial activity was time or concentration dependent and whether it differed across strains. – this belongs in a method part of the manuscript.

Table 1 Pharmacodynamic parameters estimated for all three strains and four antimicrobials – table title needs more information

The whole table is not structured per journal requirements and must be modified, it is impossible to follow. All abbreviations must be mentioned in the footnote. Names of antibiotics need to be written in full in the table columns.

Some of the main questions raised about pharmacodynamics and more specifically time or concentration dependent killing of selected antimicrobials in NG were not adequately described in relation to the obtained results.

Up until now, MIC’s are still often regarded as the best approximation for efficacy of antibiotic therapy -  I don’t believe this claim is true. Shortcomings of MIC have been known for years.

Methods:

Three Ng strains were chosen based on their MIC, ensuring a wide variation of concentrations (Table 1). – which Table 1? Is a table you are reffering to in the methods section missing? Please correct.

MIC was determined for ceftriaxone (CRO), ertapenem (ERT), fosfomycin (FOSF) and gentamicin (GENT) using the Etest method [AB bioMérieux, Sweden] according to the manufacturer’s instructions for all three strains. Obtained MIC values were used to – This line is incomplete.

Author Response

Reply referee 2

Comments and Suggestions for Authors

Thank you for giving me the opportunity to review this paper. It could be an interesting addition to the available knowledge but the way paper is written leaves much questions unanswered. Also, the whole paper needs to be edited to fit the journal requirements regarding formatting. English needs to be carefully checked.

We have rewritten introduction, results and discussion. As far as we can see in the instructions for authors, the paper would now fit the journal requirements regarding formatting, but we are ready to discuss any issue with the publisher. The complete text has now been carefully checked by Anders Boyd, who is a native English speaker and co-author of the paper.

Some specific comments:

Introduction lacks more information on why the present study was warranted. The issue of resistance in NG is not sufficiently described, the issues with available treatment, role ceftriaxone plays etc needs to be emphasized.

We have rewritten the introduction, provided more information about guidelines and issues related to ceftriaxone resistance. One major reason why the study was warranted is that we used the same antibiotics in a clinical trial, of which the results have now been published. Especially for ertapenem and fosfomycin, few PD data were available.

We established PD parameters for each antibiotic in a ceftriaxone-susceptible strain (WHO F), ceftriaxone-resistant strain (WHO X), and a non-reference clinical strain CS03307, which were used to identify whether antimicrobial activity was time or concentration dependent and whether it differed across strains. – this belongs in a method part of the manuscript.

We agree with the referee, especially with regard to the names of the strains, and changed the text accordingly.

Table 1 Pharmacodynamic parameters estimated for all three strains and four antimicrobials – table title needs more information

We have rewritten the title and expect it is now more easy to understand

The whole table is not structured per journal requirements and must be modified, it is impossible to follow. All abbreviations must be mentioned in the footnote. Names of antibiotics need to be written in full in the table columns.

We have altered the table and have now written the (full) names of the antibiotics in an additional column at the left. We expect the table is now more easy to follow.

Some of the main questions raised about pharmacodynamics and more specifically time or concentration dependent killing of selected antimicrobials in NG were not adequately described in relation to the obtained results.

We agree with the referee that it is not always easy to draw conclusions regarding mechanisms of killing (time- or concentration dependent) by this approach, since only one dose of an antibiotic is given. Some evidence exist that time-dependent antibiotics have a lower absolute ψmin , and this is what we confirmed for ceftriaxone and ertapenem, known to be time-dependent, and also for fosfomycin. Otherwise, definitive conclusions regarding time- or concentration-dependency of antibiotics should rather be drawn from models including multiple dosages of antibiotics. We made a statement on this issue in the discussion.

Up until now, MIC’s are still often regarded as the best approximation for efficacy of antibiotic therapy -  I don’t believe this claim is true. Shortcomings of MIC have been known for years.

We agree with the referee and rephrased the sentence.

Methods:

Three Ng strains were chosen based on their MIC, ensuring a wide variation of concentrations (Table 1). – which Table 1? Is a table you are reffering to in the methods section missing? Please correct.

MICs as measured by us were and are shown in the main table, however, we agree with the referee that this might have been unclear. We omitted the reference to table 1 in this section and we just mentioned the MICs that were relevant for the choice of the strains,.

MIC was determined for ceftriaxone (CRO), ertapenem (ERT), fosfomycin (FOSF) and gentamicin (GENT) using the Etest method [AB bioMérieux, Sweden] according to the manufacturer’s instructions for all three strains. Obtained MIC values were used to – This line is incomplete.

Thank you for noticing this, we have completed the sentence.

Round 2

Reviewer 1 Report

Generally, the manuscript was improved, however, it needs few additional corrections.

Introduction

Please, add in the introduction the information, that Neisseria infection treatment is becoming problematic as multidrug-resistant strains have emerged. 

Please add information what was the purpose of the research conducted.

Results

Please enlarge the graphs in Figure 1, they are unreadable in this form. Quality of this Figure needs to be improve. It's not just about resolution. Please, consider splitting the results (from Figure 1) into at least two parts and moving them to the Supplementary.

Methods:

Neisseria gonorrhoeae strains and media

Please, add the information about Neisseria cultivation in liquid media and composition of buffer, which ensures pH stabilization.

Viable cell counts

Please, complete the information, what droplets contain. It was not the rows were removed (the same comment applies Time-kill assay).  Why you consider 3- 30 colonies, and not 30 -300, which is the microbiological standard. Basing results on literally a few colonies can be misleading.

Please  consider a using appropriate kits for study the viability of bacterial cells.

Time-kill assay

0.5 McFarland of what?

Others

Literature is written in a different font type.

Reviewer 2 Report

Theee paper has improved significantly, I think it can be accepted.

Author Response

We thank the reviewer for the positive comments.